

# Conditions for super-adiabatic droplet growth after entrainment mixing

Fan Yang[1], Raymond Shaw[1], and Huiwen Xue[2]

[1]Atmospheric Sciences Program and Department of Physics, Michigan Technological University, Houghton, Michigan
[2]Department of Atmospheric and Oceanic Sciences, School of Physics, Peking University, Beijing, China

*Correspondence to:* Raymond Shaw (rashaw@mtu.edu)

**Abstract.** Cloud droplet response to entrainment and mixing between a cloud and its environment is often considered by itself, without accounting for subsequent growth after the mixing event. Here we consider the change in cloud properties when the mixed parcel rises adiabatically after the mixing event. The vertical profile for liquid water mixing ratio after a mixing event is derived analytically,

allowing the reduction due to mixing to be predicted from the mixing fraction and the cloud and environment temperature and humidity. It is derived for the limit of homogeneous mixing. The expression leads to a critical height above the mixing level: At the critical height the cloud droplet radius is the same for both mixed and unmixed parcels, and the critical height is independent of the updraft velocity and mixing fraction. Cloud droplets in a mixed parcel are larger than in an unmixed

parcel above the critical height, which we refer to as the "super-adiabatic" growth region. Analytical results are confirmed by a bin microphysics cloud model. Using the model, we explore the effects of updraft velocity, aerosol source in the environmental air, and polydisperse cloud droplets. Results show that the mixing parcel is more likely to reach the super-adiabatic growth region when the environmental air is humid and clean. It is also confirmed that the analytical predictions are matched by

the volume-mean cloud droplet radius under polydisperse conditions. The findings have implications for the origin of large cloud droplets that may contribute to onset of collision-coalescence in warm clouds.

## 1 Introduction

Warm clouds play an important role for the water cycle and energy balance in the atmosphere. How-

ever their formation, development and precipitation processes, are still not fully understood (e.g., Beard and Ochs III, 1993). Observations show that warm clouds can precipitate within 20 minutes (e.g., Laird et al., 2000; Göke et al., 2007). One open question is how small cloud droplets, which are on the order of 10 $\mu$m, change to rain drops, usually larger than 1 mm, within such a short time. Because condensation growth is slow for droplet size larger than approximately 20 $\mu$m, collision



growth is believed to be the most important mechanism for warm cloud precipitation (Pruppacher
et al., 1998).

However, collision efficiency is very low for droplets smaller than $r \approx 30 \ \mu$m due to hydrody-
namic interaction. For example, Hocking (1959) simulated two moving droplets in the Stokes flow
approximation, and found that the collision efficiency for a $r = 19 \ \mu$m droplet with smaller droplets
is mostly less than 0.1, and even for a $r = 30 \ \mu$m droplet it is mostly less than 0.5. Such low collision
efficiency suppresses the time required for drizzle and precipitation formation. Therefore, large cloud
droplets are needed to efficiently initiate precipitation. There are several hypotheses to explain the
formation of large cloud droplets: For example, the stochastic collision process itself may produce
a small number of "lucky" droplets with larger growth rates (Kostinski and Shaw, 2005). Another
possible mechanism is due to the giant cloud condensation nuclei. Observational results show that
giant and ultragiant CCNs often exist in the atmosphere, and simulation results indicate that they can
be sufficient to start the rain precipitation (e.g., Johnson, 1982; Feingold et al., 1999; Yin et al., 2000;
Blyth et al., 2003; Jensen and Lee, 2008; Cheng et al., 2009). But in this paper we focus on mech-
anisms involving the condensation process. For example, results from Lagrangian tracking studies
suggest that large droplets from condensation growth within parcels having favored trajectories can
trigger collisions and drizzle formation in warm clouds (Lasher-Trapp et al., 2005; Cooper et al.,
2013; Magaritz-Ronen et al., 2015; Naumann and Seifert, 2015; Lozar and Muessle, 2016). Korolev
et al. (2013) proposed that droplet size distribution can be broadened through diffusion growth due
to cloud base mixing and vertical fluctuation. Perhaps counter-intuitively, the mixing and entrain-
ment that occurs during cloud evolution itself may be responsible for generating large cloud droplets
(Baker et al., 1980). The possibility that that entrainment and subsequent growth can lead to droplets
larger than would occur in an unmixed parcel has occupied the attention of the cloud physics com-
munity for several decades (e.g., Baker et al., 1980; Jensen et al., 1985; Paluch and Knight, 1986;
Cooper et al., 2013; Schmeissner et al., 2015).

Observational results show that the number concentration of cloud droplets at the cloud edge/top
is usually smaller than that in the cloud due to entrainment and mixing with environmental air. How-
ever, the mean size of cloud droplet at the edge/top might be smaller, equal to, or even larger than
that in the cloud (e.g., Burnet and Brenguier, 2007; Lehmann et al., 2009; Lu et al., 2013; Beals et al.,
2015), which is thought to be the result of different mixing processes. Baker et al. (1980) proposed
two limiting mixing processes: homogeneous and extreme inhomogeneous mixing. Theoretically,
mean cloud droplet size will decrease for homogeneous mixing, but remains the same for extreme
inhomogeneous mixing. However the actual mixing process near the cloud edge/top and the response
of cloud droplets to the mixing process are still unclear. Recently, considerable theoretical and com-
putational work has been directed toward understanding the evolution of the droplet size distribution



during both homogeneous and inhomogeneous mixing processes (Andrejczuk et al., 2009; Kumar et al., 2014; Tölle and Krueger, 2014; Korolev et al., 2015; Pinsky et al., 2015b, a). Most of these analyses, however, did not consider the subsequent vertical movement of the mixed parcel, which is
also relevant to the evolution of cloud droplets (Wang and Grabowski, 2009; Yum et al., 2015; Chen et al., 2015). Finally, most theoretical work thus far does not account for the possibility of secondary activation of aerosols after dilution and mixing, although there is compelling experimental evidence that this occurs (Burnet and Brenguier, 2007; Schmeissner et al., 2015).

In this study, we are interested in the change of cloud microphysical properties after isobaric mixing of cloudy and clear-air volumes, assuming the mixing parcel rises adiabatically afterwards. This view of a single mixing event followed by isolated growth is an idealization that allows us to understand the microphysical response in the simplest of conditions. We pose the question, is it possible to achieve "super-adiabatic" droplet diameters as a result of mixing? By super-adiabatic, we mean that
the droplet diameter is larger than that observed for an unmixed, closed parcel that grows according to moist-adiabatic conditions (as defined, for example, by Cotton et al. (2011, , Chap. 4)). Specifically, we look for the conditions, such as mixing fraction, ambient humidity, aerosol entrainment, secondary activation, and vertical displacement above the mixing level, that influence the ability to produce larger droplets than exist in an unmixed parcel. We first address the problem by deriving an-
alytical results in Section 2, and then evaluate the theory and explore conditions for super-adiabatic droplet growth using a microphysical cloud parcel model in Section 3. Implications are discussed and results are summarized in Section 4.

## 2 Analytical results

As in previous studies, we consider the final state of the microphysical variables (e.g., liquid water mixing ratio, droplet sizes) after homogeneous mixing (e.g., Korolev et al., 2015). This corresponds to the limit of instantaneous mixing, under which conservation of energy and mass result in a unique dependence of droplet size on the mixing fraction (e.g., Andrejczuk et al., 2006; Burnet and Brenguier, 2007; Gerber et al., 2008; Kumar et al., 2014). Here, we consider the similar two stages of
homogeneous mixing process as discussed in (Pinsky et al., 2015b), except that the cloud parcel has continuous vertical movement after the mixing event. The first stage is (instantaneous) isobaric mixing in the absence of phase transitions, and the second stage is the response of the droplets in a vertically moving adiabatic (i.e., closed) parcel. Analytical results in this section are derived under the following assumptions: 1) only liquid exists in the condensed form (no ice); 2) the cloud par-
cel rises adiabatically; 3) the droplet size distribution is monodisperse; 4) the growth of droplets is





due to water vapor condensation; 5) sedimentation and collision–coalescence of droplets are ignored.

### 2.1 Liquid water mixing ratio in an adiabatic cloud without mixing

For reference, we begin by deriving the change of liquid water mixing ratio in a rising adiabatic cloud parcel without mixing. Considering a warm cloud parcel with monodisperse cloud droplets rising adiabatically with a constant updraft velocity, the supersaturation development equation is (Lamb and Verlinde, 2011, p. 417)

$$\frac{ds}{dt} = Q_1 w - Q_2 \frac{dq_l}{dt},$$ (1)

where $s$ is supersaturation, $w$ is updraft velocity, and $q_l$ is the liquid water mixing ratio (g kg$^{-1}$). $Q_1$ and $Q_2$ depend on temperature, pressure and other constants (all symbols and expressions are given in the Appendix). The first term on the right side represents the production of supersaturation due to adiabatic cooling due to vertical displacement, while the second term accounts for the supersaturation depletion due to vapor condensation. For monodisperse cloud droplets $q_l = (4/3)\pi\rho_w r_d^3 n_d$ where $r_d$ is the radius of cloud droplet and $n_d$ is number concentration in units of kg$^{-1}$. Thus, $dq_l/dt = 4\pi\rho_w n_d r_d^2 dr_d/dt = 4\pi\rho_w n_d r_d Gs$. Here we use the linear growth for an individual droplet: $r_d dr_d/dt = Gs$, where $G$ is the condensation growth parameter (see Appendix). By setting the production and depletion terms on the right side of Equation 1 equal to each other, we obtain the quasi-stationary supersaturation within the cloud parcel:

$$s_{qs} = \frac{Aw}{r_d n_d},$$ (2)

where $A$ is a parameter depending on $G$, $Q_1$ and $Q_2$ (see Appendix).

The linear growth rate of cloud droplets in the quasi-stationary region is

$$r_d \frac{dr_d}{dt} = Gs_{qs}.$$ (3)

Combining Equations 2, 3 and the definition of liquid water mixing ratio $q_l$ for monodisperse droplets, we can get the linear growth rate of $q_l$,

$$\frac{dq_l}{dt} = C_1 w,$$ (4)

where $C_1 = Q_1/Q_2$ with the units of m$^{-1}$ (see Appendix). If we assume $C_1$ is a constant, then $q_l$ can be derived by integration of Equation 4,

$$q_l = C_1 z + q_{l,i},$$ (5)

where $q_{l,i}$ is the initial liquid water mixing ratio, and $z = \int w dt$ is the displacement of the cloud parcel away from its initial location. The liquid water mixing ratio increases linearly with height and





does not depend on the updraft velocity. It should be mentioned that Equation 5 describes $q_l$ under thermodynamic equilibrium conditions. In reality, a cloud system needs some time (phase relaxation time) to reach the equilibrium state; For liquid clouds the phase relaxation time is usually smaller than 10 s (Korolev and Mazin, 2003).

During the adiabatic process, two physical properties of the cloud parcel will be conserved: total water mass mixing ratio and liquid water potential temperature (Gerber et al., 2008), such that

$$q_{l,i} + q_{v,i} = q_{l,f} + q_{v,f} \tag{6}$$

and

$$T_i - \frac{l_w}{c_p \epsilon} q_{l,i} = T_f - \frac{l_w}{c_p \epsilon} q_{l,f} \tag{7}$$

where $q$ is the water mass mixing ratio, $T$ is temperature, $l_w$ is the latent heat of liquid water and $c_p$ is the specific heat of air at constant pressure. Subscripts $l$ and $v$ represent liquid and water vapor, respectively, while subscripts $i$ and $f$ denote the initial and final states of the cloud parcel. We note that, for simplicity, the linearized form of the liquid water potential temperature has been used in Equation .

## 2.2 Liquid water mixing ratio in an adiabatic cloud after mixing

Now we consider the mixing of a cloud with dry and clean (aerosol free) environmental air and subsequent evolution for a closed, rising parcel. We define the mixing fraction as $\chi$, such that $\chi$ fraction of cloud air is mixed with $(1 - \chi)$ fraction of environmental air. We assume the mixing process is isobaric, and that the time scale for the mixing is much smaller than the time scale for the response of the cloud droplets during the mixing (i.e., homogeneous mixing limit). Therefore after isobaric mixing but before any phase changes, the liquid water mixing ratio should be $q_{l,im} = \chi q_{l,i}$ and the water vapor mixing ratio should be $q_{v,im} = \chi q_{v,i} + (1 - \chi) q_{v,e}$ and the temperature of the mixed parcel should be $T_{im} = \chi T_i + (1 - \chi) T_e$. The subscript $im$ represents the initial state of mixed parcel (before the evaporation of cloud droplets in the mixed parcel) and subscript $e$ denotes the state of the environmental air. After the mixing, we assume the mixed parcel rises adiabatically again with a constant updraft velocity $w_m$. For the purposes of this derivation $w_m$ is prescribed and we do not consider the actual buoyancy of the mixed parcel with respect to the environment. Similar to Equation 6 and 2.1, we have two conservation equations that allow the liquid water mixing ratio and temperature to be determined for the final state of the mixed parcel (Kumar et al., 2014), denoted by subscript $fm$:

$$\chi(q_{l,i} + q_{v,i}) + (1 - \chi) q_{v,e} = q_{l,fm} + q_{v,fm} \tag{8}$$





and

$$\chi T_i + (1-\chi)T_e - \frac{l_w \chi}{c_p \epsilon} q_{l,i} = T_{fm} - \frac{l_w}{c_p \epsilon} q_{l,fm}. \tag{9}$$

Now we ask, how does the liquid water mixing ratio in the mixed parcel ($q_{l,fm}$) change with

height above the mixing level? What is the difference of liquid water mixing ratio in the mixing

parcel ($q_{l,fm}$) compared with that in the original parcel without mixing ($q_{l,f}$) at the same height?

How does the difference ($q_{l,f} - q_{l,fm}$) change with height? To calculate this difference, we first

subtract Equation 8 from Equation 6 to get the liquid water difference for the final state,

$$q_{l,f} - q_{l,fm} = (1-\chi)(q_{l,i} + q_{v,i} - q_{v,e}) - (q_{v,f} - q_{v,fm}). \tag{10}$$

The first term on the right side is the total water mixing ratio difference between the original and new

parcel, which depends on the initial condition of the parcel ($q_{l,i}, q_{v,i}$), the environmental air ($q_{v,e}$),

and the mixing fraction $\chi$. This difference is large when $\chi$ is small and environmental air is dry. The

second term on the right side is the water vapor mixing ratio difference. The water vapor mixing

ratio can be calculated from temperature, pressure and saturation ratio: $q_v = \frac{S e_s(T) \epsilon}{p - e_s(T)}$. Therefore the

difference of water vapor mixing ratio is

$$q_{v,f} - q_{v,fm} = \frac{S_f e_s(T_f) \epsilon}{p_f - e_s(T_f)} - \frac{S_{fm} e_s(T_{fm}) \epsilon}{p_{fm} - e_s(T_{fm})}. \tag{11}$$

This equation is accurate but not simple enough to be useful. To achieve an analytical result, we

first assume $p_f \approx p_{fm}$ because both parcels are at the same height. Secondly, we ignore $e_s$ in the

denominator because $p \gg e_s$. In addition, we assume both parcels are in quasi-steady state at that

level and that the quasi-stationary supersaturation is much smaller than 1, so that the influence of the

change of $s_{qs}$ can be ignored compared with the change of $e_s(T)$ due to temperature; thus we assume

$S_{fm} \approx S_f$. The main difference in the $q_v$ arise from $e_s(T)$ due to the temperature differences. Using

the linearized form of the Clausius-Clapeyron equation, we can approximate the difference of $e_s(T)$

as

$$e_s(T_f) - e_s(T_{fm}) \approx \frac{e_s(T_f) l_w}{p_f R T_f^2}(T_f - T_{fm}). \tag{12}$$

From the above assumptions and Equation 12, we can simplify Equation 11,

$$q_{v,f} - q_{v,fm} \approx \frac{S_f e_s(T_f) l_w \epsilon}{p_f R T_f^2}(T_f - T_{fm}). \tag{13}$$

Combining Equations 10 and 13, we find that the liquid water mixing ratio difference depends on

the temperature difference in this way,

$$q_{l,f} - q_{l,fm} = (1-\chi)(q_{l,i} + q_{v,i} - q_{v,e}) - \frac{S_f e_s(T_f) l_w \epsilon}{R T_f^2}(T_f - T_{fm}). \tag{14}$$





In addition, the difference in liquid water potential temperature conservation equations for closed
and mixed parcels given by Equation 2.1 minus Equation 9, leads to

$$(1-\chi)(T_i - T_e - \frac{l_w}{c_p \epsilon} q_{l,i}) = T_f - T_{fm} - \frac{l_w}{c_p \epsilon}(q_{l,f} - q_{l,fm}). \tag{15}$$

Finally, from Equations 14 and 15, we can obtain the approximate solutions for liquid water mixing
ratio difference and temperature difference,

$$q_{l,f} - q_{l,fm} = (1-\chi)\frac{(1+C_3)q_{l,i} + q_{v,i} - q_{v,e} - C_2(T_i - T_e)}{1 + C_3} \tag{16}$$

and

$$T_f - T_{fm} = (1-\chi)\frac{C_2(T_i - T_e) + C_3(q_{v,i} - q_{v,e})}{C_2(1 + C_3)}. \tag{17}$$

Finally, combining Equations 5 and 16, we can get the liquid water profile for the mixed parcel,

$$q_{l,fm}(z) = C_1 z + q_{l,i} - (1-\chi)K_1, \tag{18}$$

where $K_1 = ((1+C_3)q_{l,i} + q_{v,i} - q_{v,e} - C_2(T_i - T_e))/(1+C_3)$. It is interesting to see that the liquid
water mixing ratio for the mixed parcel still increases linearly with height, but with a smaller value
compared with an unmixed parcel (cf. Equation 5). The difference is the same at different heights,
and is proportional to $1 - \chi$.


### 2.2.1   Total evaporation and reactivation height

Another way to look at Equation 18 is that the liquid water mixing ratio in the mixing parcel $q_{l,fm}$
increases with height linearly with the same slope as $q_{l,f}$ in the original parcel, but with a smaller ini-
tial liquid water mixing ratio in the mixing parcel $q_{l,im} = q_{l,i} - (1-\chi)K_1$. Although the initial liquid
water mixing ratio $q_{l,im}$ should be non-negative physically, $q_{l,i} - (1-\chi)K_1$ can be negative mathe-
matically. If $q_{l,im}$ is negative, it means that all cloud droplets evaporate. Therefore, $q_{l,i} = (1-\chi)K_1$
is the criteria for critical condition that all droplets totally evaporate and the air in mixing parcel is
just saturated. It is not difficult to prove that this critical condition is consistent with that given by
Pinsky et al. (2015b), with $\gamma = 0$.


Even if $q_{l,fm}$ is negative at $z = 0$, it can become positive at higher altitude. The negative value
of $q_{l,fm}$ at the beginning is the result of total evaporation, while the point where $q_{l,fm}$ changes
to positive can be taken to represent the re-activation of cloud condensation nuclei to form cloud
droplets. The re-activation height $z_{react}$ is the distance between the mixing level and the level at
which $q_{l,fm} = 0$, given by

$$z_{react} = \frac{(1-\chi)K_1 - q_{l,i}}{C_1}. \tag{19}$$





### 2.2.2 Critical height for superadiabatic droplet growth

In this subsection we consider how cloud droplet size changes with height above the mixing level. We consider an initially-adiabatic cloud parcel mixed isobarically with clean environmental air at some level above the cloud base. Without vertical movement, the liquid water mixing ratio and cloud number concentration will decrease due to dilution (not considering, for the moment, scenarios in which all droplets are evaporated). The mean cloud droplet size after the response to mixing is

the same for extremely inhomogeneous mixing, but smaller for homogeneous mixing. If the parcel still rises adiabatically after mixing, however, the liquid water mixing ratio will increase with height (cf. Equation 18). This indicates that cloud droplet size will also increase with height, because the number concentration does not change during the vertical motion. We now consider the growth of cloud droplets under quasi-steady conditions. Because the cloud droplet concentration is smaller in

the mixed parcel than in the original parcel, $s_{qs}$ in the mixed parcel will be larger ($s_{qs} \propto (r_d n_d)^{-1}$, see Equation 2). This implies that cloud droplets in the mixing parcel grow faster than those in the original one due to higher supersaturation. This suggests that although cloud droplet size in the mixed parcel is smaller for homogeneous mixing at the beginning, it might, with adequate vertical displacement, become equal to or even larger than that in the original, unmixed parcel. The resulting

droplets would appear to have experienced super-adiabatic growth compared to a closed parcel. This general picture of large-drop production resulting from decreased competition in diluted parcels has been discussed elsewhere in the literature (Paluch and Knight, 1986; Cooper et al., 2013; Schmeissner et al., 2015).

We define the critical height $z^*$ as the height when droplets in both parcels have the same sizes. Based on the definition of liquid water mixing ratio, it is apparent that $q_{l,fm}/q_{l,f} = \chi$ at the critical height, and therefore

$$\frac{C_1 z^* + q_{l,i} - (1-\chi)K_1}{C_1 z^* + q_{l,i}} = \chi. \tag{20}$$

Solving Equation 20, we obtain

$z^* = \dfrac{K_1 - q_{l,i}}{C_1}. \tag{21}$

We note with interest that $z^*$ is independent of the mixing fraction $\chi$. Equations 18 and 21 indicate that although the liquid water mixing ratio for the mixed parcel is always lower than that in the original parcel, droplet radius in the mixed parcel will be larger than that in the original parcel when the parcel is above $z^*$.




## 3   Results from parcel model

The analytical results derived in Section 2 have provided insight into the evolution of a cloud parcel after a mixing event, but several assumptions and simplifications were made, and some limitations such as perfectly clean (aerosol free) environment were imposed. We now explore the same con-

cept of idealized mixing and subsequent-growth, but using an adiabatic parcel model with bin microphysics. The model was originally developed by Feingold et al. (1998) to simulate warm cloud process and has been applied to a wide range of microphysical problems (Feingold and Kreidenweis, 2000; Xue and Feingold, 2004; Ervens et al., 2005; Ervens and Feingold, 2012; Yang et al., 2012; Li et al., 2013). To study the mixing process, we add an idealized entrainment/detrainment and mix-

ing process to the model. Entrainment means some fraction of environment air flows into the cloud, while detrainment means some fraction of cloud flows into the environment (de Rooy et al., 2013). The mixing process is implemented so that the entrained environmental air is homogeneously mixed with the remaining cloud body, and in all cases considered here this mixing level is set to 665 m (50 m above cloud base). It should be mentioned that mixing process might not necessarily happen when

entrainment/detrainment occurs. The time interval between these two processes is called the mixing time scale, and the presence of a delay would be expected for inhomogeneous mixing. The relative magnitudes of this mixing time scale and the phase relaxation time determine whether the mixing occurs in the homogeneous or inhomogeneous limit (e.g., Baker et al., 1980). To be consistent with the previous theoretical development (Sec. 2) we implement the homogeneous mixing limit within

the model, i.e., the instantaneous exposure of droplets to the mixture of cloudy and entrained air. This implies that the turbulent mixing time is very small compared to the phase relaxation time, and is therefore similar to the limit considered by Pinsky et al. (2015b).

Initial conditions for the parcel are $z_0 = 300$ m, $p_0 = 919$ Pa, $T_0 = 288.15$ K and $RH_0 = 85\%$.

Cloud condensation nuclei (CCN) are ammonium sulfate particles with a monodisperse radius of 50 nm and concentration of 50 mg$^{-1}$. The parcel rises adiabatically with constant updraft velocity. Two updraft velocities ($w$) are chosen in this study: 0.1 and 1.0 m s$^{-1}$. Observation results show that updraft velocity in cumulus cloud is on the order of 1.0 m s$^{-1}$, and that for stratocumulus cloud is on the order of 0.1 m s$^{-1}$ (Katzwinkel et al., 2014; Ditas et al., 2012). Cloud base is reached at $z = 615$

m, where CCN are activated as cloud droplets. The isobaric mixing process occurs at $z = 665$ m, 50 m above the cloud base. For simplicity, we assume the environmental temperature at the mixing level is the same as that of the cloud parcel, but the relative humidity is only $85\%$. After the mixing, the new mixed parcel rises adiabatically again with the same updraft velocity.

Liquid water mixing ratio profiles for six different mixing fractions $\chi = 1.0, 0.9, 0.8, 0.7, 0.6, 0.5$ at $w = 0.1$ m s$^{-1}$ are shown in Figure 1 (a). The analytical results based on Equation 18 are also shown and are quite close to the results from the parcel model. As seen from Figure 1 (a), the liquid





water mixing ratio for smaller $\chi$ is smaller than that for larger $\chi$ at the same height. In addition, when $\chi \leq 0.8$, the liquid water mixing ratio will decrease to zero at the beginning, which means that the

cloud totally evaporates and becomes subsaturated. It should be mentioned that in the model each cloud droplet contains one CCN, and when a cloud droplet totally evaporates the CCN still survives. Because the subsaturated parcel still rises adiabatically, CCN in the mixing parcel can be activated again when the air becomes saturated at a higher level, which we defined as the re-activation level. The smaller $\chi$ is, the higher the re-activation level is. The evaporation and re-activation processes can

be clearly seen from the cloud droplet radius profile in Figure 1 (b). In addition, it clearly shows that the mixed cloud parcel can reach super-adiabatic growth conditions (where the cloud droplet radius in the mixed parcel is larger than that in the original, unmixed parcel with $\chi = 1.0$) above a critical height. The critical height is independent of $\chi$ and agrees well with that predicted by Equation 21.

Results above are for a cloud parcel mixing with clean environmental air (aerosol free condition). However, both observational and modeling results show that air around the cumulus cloud is usually not clean(Katzwinkel et al., 2014; Chen et al., 2012). There can be background aerosols in the atmosphere even at high altitude, and in addition, subsiding shells can also provide sufficient aerosols as CCN when mixing occurs (Heus and Jonker, 2008). There is no simple analytical result for mixing

with a polluted environment. However, we can use the parcel model to investigate the effect of mixing when the environmental air is polluted. For simplicity, we assume the environment has the same dry aerosol size distribution as that below the cloud base.

Figure 1 (c) shows the monodisperse cloud droplet radius versus height for various $\chi$ after mixing

with a polluted environment at $w = 0.1$ m s$^{-1}$. For $\chi = 0.9$, the remaining cloud droplets do not totally evaporate and the entrained aerosols are not activated as cloud droplets. For smaller $\chi$, the remaining cloud droplets totally evaporate and leave CCN in the mixed parcel. Both entrained and remaining CCN are activated at a higher level. In addition, only the parcel with $\chi = 0.9$ can reach the super-adiabatic growth region. For smaller $\chi$, cloud droplets are smaller than those in the original

parcel at the same height $z^*$. This is similar to the aerosol indirect effect in which more aerosols leads to smaller cloud droplets. In summary, when mixing with a polluted environment, the mixing parcel can reach super-adiabatic growth conditions at the predicted $z^*$ only if the cloud does not totally evaporate after mixing.

Figure 2 (a) and (b) show the results for mixing with a clean environment at larger updraft velocity $w = 1.0$ m s$^{-1}$. It can be seen that the liquid water mixing ratio and cloud droplet radius profiles are almost the same compared with Figure 1, except that the mixing parcel totally evaporate for $\chi = 0.8$ at $w = 0.1$ m s$^{-1}$, but doesn't totally evaporate for $\chi = 0.8$ at $w = 1.0$ m s$^{-1}$. This is because larger updraft velocity supplies more water within the fixed phase relaxation time, so droplets begin to grow



before they have had time to completely evaporate. The mixed parcel can reach the super-adiabatic growth region when it is above $z^*$. And as before, $z^*$ is independent of both mixing fraction and updraft velocity, consistent with the theoretical prediction.

When mixing with polluted environment air at $w = 1.0 \text{ m s}^{-1}$, the mixed parcel can't reach the

super-adiabatic growth region whether the mixing parcel totally evaporates or not (see Figure 2 (c)). The reason is that with large updraft velocity, the entrained CCN can always be activated as cloud droplets, thus compete for water vapor in the mixed parcel. It should be mentioned that results here strongly depend on the physical and chemical properties of the entrained CCN, e.g, sizes, chemical composition, and number concentration. For example, the mixed parcel might also reach the super-

adiabatic growth region if the environmental air only contains a small number of CCN. In general, however, mixing with polluted air will inhibit the super-adiabatic growth of cloud droplets.

Cloud droplets in a real cloud are usually polydisperse and we now consider to what extent the theoretical predictions apply in this more complex system. The effect of mixing on a polydisperse

droplet population is tested with the cloud parcel model. The initial aerosols are composed of ammonium sulfate and are distributed lognormally in 20 bins with 50 nm median radius, standard deviation of 1.4, and a total number concentration of $100 \text{ cm}^{-3}$. Initial radii of the dry aerosols for the 20 bins are listed in Table 2. The cloud droplet diameters for each bin versus height for $\chi = 0.9, 0.7, 0.5$ are shown in Figure 3. These results are for clean environmental air and $w = 0.1 \text{ m s}^{-1}$ and are represen-

tative of the other cases. It can be seen that not all 20 bins are activated at cloud base; for example, only the largest 11 aerosol sizes are activated as cloud droplets for $\chi = 1.0$. Cloud droplets evaporate a little bit for $\chi = 0.9$, or completely for $\chi = 0.7, 0.5$, and re-activation occurs again at a higher level. It is very interesting to see that for $\chi = 0.5$, the 12th bin is not activated at cloud base, but is activated for the first time after mixing (green line). This asymmetric phenomenon is due to the sig-

nificant reduction of cloud droplet number concentration after mixing. Thermodynamic equilibrium predicts how much water vapor should condense at a certain level, but mixing with a clean environment reduces the overall CCN concentration. To condense the same amount of water, either the single droplets must grow larger than before, which is the physical explanation for super-adiabatic growth; or some initially un-activated aerosol particles can be activated to increase the cloud number

concentration.

Super-adiabatic droplet growth for individual droplet size bins can be observed in Figure 3, but it is achieved at different heights above the mixing level. Figure 4 shows these critical heights for individual cloud droplet size bins calculated from the cloud parcel model for the various mixing

fractions. Here again, the environmental air is clean with $T_e = T_c$ and $RH_e = 85\%$. We note that cloud droplet size decreases with increasing bin number (i.e., cloud droplet size increases with in-





creasing dry aerosol size, as expected). The critical height for each bin is defined when the sizes
of cloud droplets for that bin are equal for both mixed and unmixed cloud parcels. It can be seen
that the critical height depends on the size of the cloud droplet, the mixing fraction and the updraft
velocity, especially for low updraft velocity $w = 0.1$ m s$^{-1}$. For $w = 1.0$ m s$^{-1}$, critical heights for
individual bins are close to the theoretical critical height for a monodisperse cloud droplet popula-
tion. In the low updraft speed case (left panel) it is particularly striking that the $\chi = 0.9$ curve has
a different dependence than that for the other mixing fractions: increasing rather than decreasing $z^*$
with decreasing droplet size. We believe the explanation is that the $\chi = 0.9$ case is the only scenario
in which complete droplet evaporation does not occur. Thus, the presence of complete evaporation
and subsequent re-activation changes the population dynamics of the cloud substantially for low up-
draft speeds. Although the critical heights are different for individual size bins, we might expect that
the simple monodisperse prediction for $z^*$ would hold for some moment of cloud droplet size distri-
bution. Considering that the thermodynamically-predicted water mass is distributed over a variable
number of aerosol particles, the most logical choice is a prediction of $z^*$ using the volume-mean
radius. Figure 5 shows the volume-weighted mean radius as a function of height for six values of $\chi$
and for updraft speeds of $0.1$ and $1.0$ m s$^{-1}$. In spite of the complex behavior observed for individual
bins, the volume-mean radius curves are observed to cross at nearly the same height and with very
close agreement with the analytical prediction. This suggests that the theory can be applied under
realistic cloud conditions with polydisperse droplet populations.

## 4   Discussions and conclusions

In this study, we have considered isobaric mixing of a cloud parcel with environmental air, and then
the subsequent droplet growth as the parcel rises adabatically afterwards. Analytical expressions are
derived for monodisperse cloud droplets when mixing with clean environmental air. Results show
that the liquid water mixing ratio $q_l$ in the mixed parcel increases linearly with height with the same
slope ($\frac{dq_l}{dz}$) as the original parcel (without mixing). Due to the mixing the $q_l$ is smaller compared with
the unmixed parcel at the same height. A closed form expression for the offset is derived and shows
that the decrease of $q_l$ in the mixed parcel depends on the mixing fraction $\chi$ and the temperature
and relative humidity of the environmental air. A critical height $z^*$, defined as the height at which
the cloud droplet sizes are equal in both mixed and original cloud parcels, is derived. Interestingly,
the critical height depends on the initial conditions of the cloud and environmental air, but is inde-
pendent of the mixing fraction. Cloud droplets in the mixed parcel are larger than in the original
parcel above $z^*$, which we call the "super-adiabatic" growth region. These large cloud droplets may
help explain the formation of initial large droplets that contribute to precipitation formation in warm



clouds.

The predicted vertical profile of liquid water mixing ratio and the critical height are confirmed using a bin microphysical cloud model. For large $\chi$ and a humid environment, cloud droplets will
evaporate a little bit and grow again after mixing. For small $\chi$ and dry environment, cloud droplets can evaporate completely, leaving the mixed parcel subsaturated. Droplets are re-activated at a higher level, as long as the mixing parcel rises sufficiently to reach saturation again. The theoretical predictions are based on equilibrium arguments, but because the phase relaxation time is typically short for warm clouds, results are not very sensitive to updraft speed over the range investigated. For monodis-
perse cloud droplets, $z^*$ is independent of mixing fraction and updraft speed. For polydisperse cloud droplets, however, $z^*$ defined for individual droplet sizes is observed to depend on droplet size, mixing fraction and updraft velocity, especially for smaller $w$. For larger $w$, $z^*$ is insensitive to those variables and close to the analytical result for monodisperse cloud droplets. The situation becomes much simpler and the polydisperse cloud can be predicted theoretically by using the volume-mean
cloud droplet radius. Finally, we note that the model results presented here are for the condition of cloud and environment having the same temperature; model runs for other reasonable conditions also produced good agreement with the theory.

Environment background aerosols and subsiding shells may contain effective CCN that can be
activated after mixing, thus inhibiting super-adiabatic droplet growth. For large updraft speed, the entrained aerosols can be activated as cloud droplets, thus increasing cloud droplet concentration and decreasing the cloud droplet sizes. For small updraft velocity, the mixed parcel can reach the super-adiabatic growth region only when the entrained aerosols cannot be activated and the cloud droplets do not totally evaporate. Otherwise if cloud droplets totally evaporate, both remaining and
entrained CCN can be activated when the mixed parcel is saturated again. If the entrained aerosols can be activated as cloud droplets, the mixed parcel usually contains smaller cloud droplets, but similar number concentration compared with the main cloud body. This might help explain the observation that some cloud samples appear to be undiluted in droplet number concentration, but have significantly smaller mean-volume radii, a region otherwise outside the homogeneous mixing limit-
ing curve in a mixing diagram (Schmeissner et al., 2015).

Given the success of the analytical results in predicting the critical height $z^*$ above which volume-weighted mean droplet diameters will appear to be super-adiabatic, we briefly explore the dependence of $z^*$ on environmental conditions. As noted already, and now confirmed by the parcel model,
the critical height does not depend on mixing fraction $\chi$ or on the updraft speed $w$. As seen in Figure 6, $z^*$ changes with the relative humidity of the environmental air ($RH_e$) at the mixing level. It can be seen that $z^*$ decreases as $RH_e$ increases. For example, when $RH_e = 98\%$, $z^*$ is less than 50



m above the mixing level. This means that the mixed parcel can reach the super-adiabatic growth region more easily when mixing with a humid environment. In the real atmosphere, the subsiding

shell around a cumulus cloud in a clean environment might be very humid due to the evaporation of cloud droplets at higher cloud levels. Mixing under these conditions would be favorable for super-adiabatic growth of cloud droplets, especially if the subsiding shell has been cleared for most CCN through scavenging.

The results presented here all are for the homogeneous mixing limit. It is possible to develop model prescriptions for extreme inhomogeneous mixing, but our sense is that the results would be sensitive to the necessarily artificial nature of those prescriptions. Ultimately, a realistic model or a direct numerical simulation of the mixing process are required for the inhomogeneous limit. We can speculate, however, that the results obtained here would only be amplified for inhomogeneous

mixing: in that limit the droplet concentration is reduced but the mean volume diameter remains unchanged, implying that $z^*$ is zero and super-adiabatic droplet growth can begin immediately after the mixing process has concluded. By concluded we mean that the cloudy and environmental air have become completely mixed, leaving a spatially homogeneous field of droplets having the same diameter as before mixing, but lower number concentration due to dilution and total evaporation of

some subset of droplets (e.g., Beals et al., 2015). This neglects the more complicated interactions that might come into play if CCN are entrained during mixing with environmental air: in that case activation of new CCN may occur as the parcel rises, even before complete mixing to the microscale has taken place.

A crucial factor that has not been considered thus far is the influence of mixing on the vertical mo-

tion of a cloud parcel due to changes in its buoyancy. Whether a mixed cloud parcel can experience super-adiabatic droplet growth depends not only on the critical height $z^*$, but also on the maximum height $z_{max}$ it can reach after mixing: a cloud can reach the super-adiabtic growth region only for $z_{max} > z^*$. Calculation of $z_{max}$ is nontrivial because one must consider the time dependence of the buoyancy, drag force, and kinetic energy of the parcel, which depends on the properties of the

surrounding environment and and its dependence on height. These are still open research problems (e.g., slippery versus sticky thermals (Sherwood et al., 2013; Romps and Charn, 2015)), so exploring this important aspect is beyond the scope of our paper; but qualitatively, our results imply that strongly convective clouds may favor super-adiabatic growth compared to weakly convective clouds. In addition, decreasing $\chi$ will tend to decrease the buoyancy and therefore the updraft speed, thus

ultimately decreasing $z_{max}$. Therefore, it is more likely to reach the super-adiabatic droplet growth region for larger $\chi$, again favoring clouds in humid environments or clouds with well developed, humid subsiding shells.





In a real cloud the liquid water mixing ratio profile is much more complicated than considered
here. Mixing will occur at different levels and environmental conditions change with height. There
are several methods to predict the mixing fraction at different levels. For example, Lu et al. (2012)
predict $\chi$ using the cloud base condition, liquid water mixing ratio and environmental condition at
each level. The advantage of their method is that they do not need to measure temperature and water
vapor mixing ratio in the cloud, which have significant measurement uncertainty. Here, we have pro-
vided an explicit method to estimate the mixing fraction at each level using a similar strategy. Based
on Equation 16 and 17, we can also calculate the mixing fraction profile. The key difference is that
our method is explicit, while their method is implicit.

The central insights of this work are that a mixed parcel is more likely to reach the super-adiabatic
growth region when convection is strong, and the environmental air is humid and clean. Cloud
droplets in the super-adiabatic growth region are larger than that in an unmixed parcel. Our hope
is that the theoretical results obtained here and confirmed with the parcel model, can help in evaluat-
ing the possible role of mixing-induced droplet growth for large droplet production and development
of precipitation in warm clouds.

*Acknowledgements.* This research was supported by the DOE Office of Science as part of the Atmospheric
System Research program through Grant No. DE-SC0011690.



## Appendix A: List of Symbols

**Table 1.** List of Symbols

| Symbol | Description | Units |
|---|---|---|
| $A$ | $\frac{Q_1}{4\pi\rho_w G Q_2}$ | s kg$^{-1}$ |
| $c_p$ | specific heat of air at constant temperature | J mol$^{-1}$ |
| $C_1$ | $4\pi\rho_w G A = Q_1/Q_2$ | m$^{-1}$ |
| $C_2$ | $\frac{S_f e_s(T_f) l_w \epsilon}{p_f R T_f^2}$ | K$^{-1}$ |
| $C_3$ | $\frac{C_2 l_w}{c_p \epsilon}$ | — |
| $D_v$ | Diffusivity of water vapor | m$^2$ s$^{-1}$ |
| $e_v$ | water vapor pressure | Pa |
| $e_s(T)$ | saturated water vapor pressure at temperature $T$ | Pa |
| $G$ | $\left[\frac{\rho_w RT}{M_w D_v e_s(T)} + \frac{\rho_w l_w}{M_w k_T T}\left(\frac{l_w}{RT}-1\right)S\right]^{-1}$ | m$^2$ s$^{-1}$ |
| $k_T$ | coefficient of air heat conductivity | J m$^{-1}$ s$^{-1}$ |
| $K_1$ | $\frac{(1+C_3)q_{l,i}+q_{v,i}-q_{v,e}-C_2(T_i-T_e)}{1+C_3}$ | — |
| $K_2$ | $\frac{C_2(T_i-T_e)+C_3(q_{v,i}-q_{v,e})}{C_2(1+C_3)}$ | K |
| $l_w$ | latent heat of liquid water | J mol$^{-1}$ |
| $M_{air}$ | molar mass of air | kg mol$^{-1}$ |
| $M_w$ | molar mass of water | kg mol$^{-1}$ |
| $n_d$ | droplet number per unit mass of air | kg$^{-1}$ |
| $q_l$ | liquid water mixing ratio | — |
| $q_{l,i}$ | initial $q_l$ | — |
| $q_{l,im}$ | initial $q_l$ with mixing fraction $\chi$ | — |
| $q_{l,f}$ | final $q_l$ | — |
| $q_{l,fm}$ | final $q_l$ with mixing fraction $\chi$ | — |
| $q_v$ | water vapor mixing ratio | — |
| $q_{v,e}$ | environmental $q_v$ | — |
| $q_{v,i}$ | initial $q_v$ | — |
| $q_{v,im}$ | initial $q_v$ with mixing fraction $\chi$ | — |
| $q_{v,f}$ | final $q_v$ | — |
| $q_{v,fm}$ | final $q_v$ with mixing fraction $\chi$ | — |
| $Q_1$ | $(l_w/(c_p T)-1)(M_{air}g/RT)$ | m$^{-1}$ |
| $Q_2$ | $\rho_{air}l_w^2/(M_w p c_p T)+\rho_{air}RT/(M_w e_s(T))$ | — |
| $r_d$ | radius of cloud droplet | m |
| $r_{d,i}$ | initial $r_d$ | m |
| $r_{d,f}$ | final $r_d$ | m |
| $r_{d,fm}$ | final $r_d$ with mixing fraction $\chi$ | m |
| $R$ | universal gas constant | J mol$^{-1}$ K$^{-1}$ |
| $s$ | $S-1$, water vapor supersaturation | — |
| $S$ | $\frac{e_v}{e_s}$, water vapor saturation ratio | — |
| $S_f$ | final $S$ | — |
| $S_{fm}$ | final $S$ with mixing fraction $\chi$ | — |
| $T$ | temperature | K |
| $T_i$ | initial $T$ | K |
| $T_{im}$ | initial $T$ with mixing fraction $\chi$ | K |
| $T_e$ | environmental $T$ | K |
| $T_f$ | final $T$ | K |
| $T_{fm}$ | final $T$ with mixing fraction $\chi$ | K |
| $w$ | updraft velocity of cloud parcel | m s$^{-1}$ |
| $w_m$ | updraft velocity of cloud parcel with mixing fraction $\chi$ | m s$^{-1}$ |
| $\chi$ | isobaric mixing fraction | — |
| $\epsilon$ | $\frac{M_w}{M_{air}}$ | — |
| $\kappa$ | $\frac{R}{c_p}$ | — |
| $\rho_w$ | density of liquid water | kg m$^{-3}$ |
| $\rho_{air}$ | density of air | kg m$^{-3}$ |



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





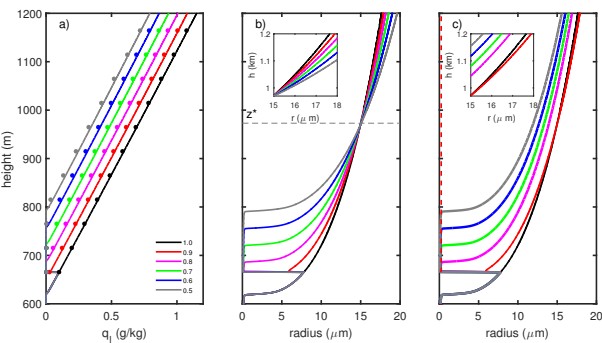

**Figure 1.** (a) Liquid water mixing ratio profiles for various cloud mixing fractions $\chi$ and with low updraft speed (0.1 m s$^{-1}$). Lines are from the parcel model and dots are from the theoretical prediction given by Equation 18. (b) Cloud droplet radius versus height for various $\chi$ when mixing with clean (aerosol free) environmental air. The horizontal dashed line represents the critical height $z^*$ calculated from Equation 21. (c) Cloud droplet radius versus height for various $\chi$ when mixing with polluted environmental air (air containing CCN similar to cloud base conditions). Insets in (b) and (c) show details of the radius profiles above the critical height. Super-adiabatic droplet growth, i.e. droplet diameters greater than in the unmixed cloud ($\chi = 1.0$), is observed for all $\chi$ in (b) and only for $\chi = 0.9$ in (c).

**Table 2.** Initial dry aerosols radii for different bins

| Bin number | $r_{dry}$ (nm) | Bin number | $r_{dry}$ (nm) |
|---|---|---|---|
| 1 | 463.7 | 11 | 61.5 |
| 2 | 378.9 | 12 | 50.3 |
| 3 | 309.6 | 13 | 41.1 |
| 4 | 253.0 | 14 | 33.6 |
| 5 | 206.7 | 15 | 27.4 |
| 6 | 168.9 | 16 | 22.4 |
| 7 | 138.0 | 17 | 18.3 |
| 8 | 112.8 | 18 | 15.0 |
| 9 | 92.1 | 19 | 12.2 |
| 10 | 75.3 | 20 | 10.0 |





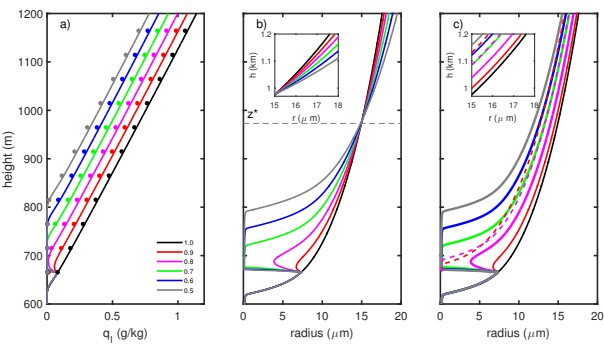

**Figure 2.** (a) Liquid water mixing ratio profiles for various cloud mixing fractions $\chi$ and with high updraft speed (1.0 m s$^{-1}$). Lines are from the parcel model and dots are from the theoretical prediction given by Equation 18. (b) Cloud droplet radius versus height for various $\chi$ when mixing with clean (aerosol free) environmental air. The horizontal dashed line represents the critical height $z^*$ calculated from Equation 21. (c) Cloud droplet radius versus height for various $\chi$ when mixing with polluted environmental air (air containing CCN similar to cloud base conditions). Insets in (b) and (c) show details of the radius profiles above the critical height. Super-adiabatic droplet growth, i.e. droplet diameters greater than in the unmixed cloud ($\chi = 1.0$), is observed for all $\chi$ in (b) but for none in (c).





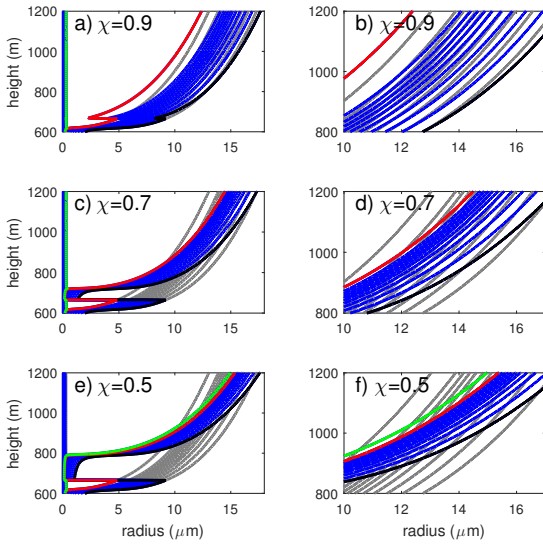

**Figure 3.** Radii of cloud droplets in a polydisperse population versus height for $\chi = 0.9, 0.7, 0.5$ in a clean environment at $w = 0.1$ m s$^{-1}$. The background grey lines represent $\chi = 1.0$. The right column shows the region near the critical height where super-adiabatic growth can be expected. The black line is for the 1st bin (largest CCN), the red line for the 11th bin, and the green line for the 12th bin.

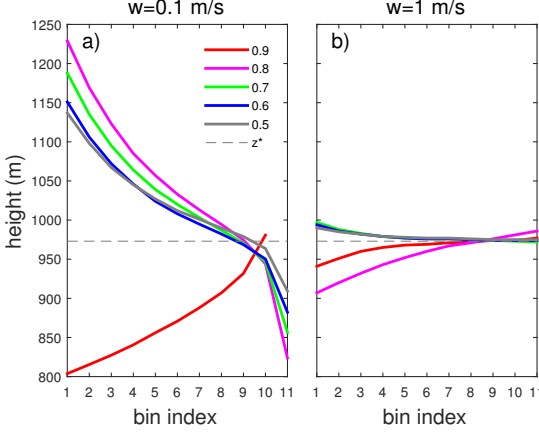

**Figure 4.** Critical height for individual droplet size bins for a polydisperse cloud droplet population calculated from cloud parcel model. Results are shown for two updraft velocities, (a) $w = 0.1$ m s$^{-1}$ and (b) $w = 1.0$ m s$^{-1}$. The line colors represent different mixing fractions $\chi$ as defined in the legend, and the dashed line is the analytical result for critical height $z^*$ for a monodisperse cloud droplet population. Cloud droplet size decreases as the bin number increases.





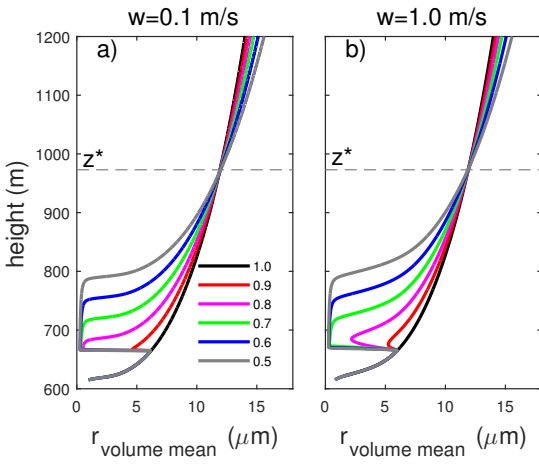

**Figure 5.** Volume-mean radius for a polydisperse cloud droplet population versus height at updraft speeds of a) $w = 0.1 \, \mathrm{m \, s^{-1}}$ and b) $w = 1.0 \, \mathrm{m \, s^{-1}}$ and for a clean environment. Line colors represent different mixing fractions $\chi$, as in Figures 1 and 2. The horizontal dashed line is the critical height $z^*$ predicted for a monodisperse cloud droplet population with equal volume-mean radius.

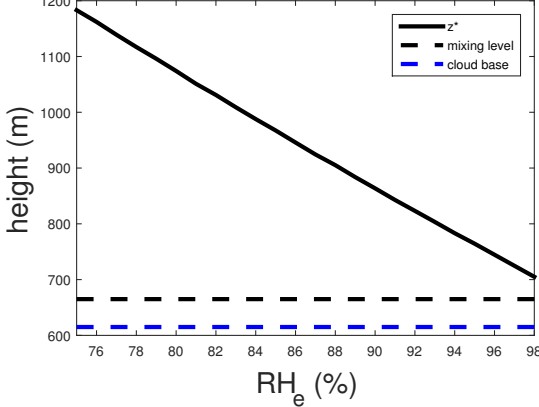

**Figure 6.** Critical height $z^*$ versus environmental relative humidity $RH_e$ at the mixing level. The height of cloud base (blue dashed line) and the mixing level (black dashed line) are shown for reference.