# Peer review of "Conditions for super-adiabatic droplet growth after entrainment mixing"

_Atmospheric Chemistry and Physics, 2016_

## Referee Comment (RC1) · Anonymous Referee #1 · 4 Mar 2016

This paper investigates if cloud parcels that go through a mixing event can produce larger droplets than undisturbed parcels (this is called super-adiabatic growth by the authors). Mixed parcels contain less water and fewer droplets than undisturbed parcels, and therefore droplets there grow faster after a mixing event, although starting from a smaller radius. From thermodynamical considerations, the authors show that super-adiabatic growth is expected for a pristine environment when mixed parcels rise to a given height. This height mostly depends on the thermodynamical properties of the cloud and of the environment, and it is independent of the updraft velocity and of the mixing fraction. This result is tested with a parcel model for different updrafts velocities, different polluted environments and for a polydisperse droplet population.

It has been argued in the past that super-adiabatic droplet growth can help to explain rain formation in warm clouds. The authors are able to quantify this effect in idealized

conditions, and I think that their results can be used to estimate the relevance of the mechanism in future studies. For these reasons, I think that the work can be a worthy publication for ACP if the authors answer the next questions.

1) One of the inherent assumptions for the parcel model and for the thermodynamical calculations is that the parcel only mixes once with the cloud-free environment, which means that it never mixes with cloudy air. I would like that the authors discuss this assumption more in detail. Clouds are turbulent and continuously mix (see for example Margaritz et al. 2014), which homogenizes the droplet number concentration. In the example from Figure 1, the parcel has to rise for ∼3000 seconds without mixing with other cloud parcels in order to become super adiabatic. It seems unlikely for me to find such a parcel in a real cloud.

2) It would be interesting to know the authors conclusions about the role of super adiabatic droplets for large droplets production and rain formation, from the results presented in the paper. Do they think that the mechanism is relevant for all warm clouds or only in a few particular cases (very wet and very clean environment)? Do they think that the mechanism is relevant for stratocumulus (which are usually thin (∼300 m), with a dry capping free atmosphere and with mixing only on the top)? Can they estimate how does the droplet size distribution broaden due to this mechanism (is it sufficient for rain formation)?

Technical corrections:

1) Line 120. Equation (4) can be directly obtained from Eq. (1) in the quasi-stationary limit. No need to refer to Eqs. (2) and (3).

2) Line 135. There is a prefactor missing in the definition of the liquid potential temperature, which accounts for the pressure dependence. With the current definition, liquid potential temperature is only conserved for adiabatic and isobaric processes. Also, I do not see why \epsilon appears in the definition (it does not appear in Gerber et al. 2008).
3) Line 141. Equation number missing.

4) Line 151. I would not discuss the state im (after mixing but before phase changes). It is not very useful for the discussion, and it adds more symbols.

5) Line 156 and 192. Equation 2.1 does not exist.

6) Line 246. Explain better why ql,fm /ql,f = \chi is the condition for the critical height. Remind the condition of completely clean environment.

7) Line 316. Provide the number of CCN in the polluted environment.

References: Magaritz-Ronen, L., Pinsky, M., and Khain, A.: Effects of Turbulent Mixing on the Structure and Macroscopic Properties of Stratocumulus Clouds Demonstrated by a Lagrangian Trajectory Model, J. Atmos. Sci., 71,1843–1862, 2014.

---

## Referee Comment (RC2) · Anonymous Referee #2 · 9 Mar 2016

Review of ACP-2016-94

In this manuscript the authors tried to tackle the problem of super-adiabatic droplet growth, which has been a subject of great interest in cloud physics community for the last several decades. For without such growth, warm rain initiation within a realistic time scale seems very difficult, if not impossible. The authors considered entrainment and mixing processes as a key to the super-adiabatic droplet growth and derived equations that could calculate analytically the variation of temperature and liquid water mixing ratio and thus droplet radius after entrainment and mixing and during the further ascent of the mixed cloud parcel. Then the authors demonstrated that this theoretical formulation was consistent with the results of cloud parcel model simulations of such processes. Moreover, the authors suggested some proper environmental conditions for super-adiabatic droplet growth. This manuscript does add some new insights on

super-adiabatic droplet growth and could be worth publication in ACP if the following issues are handled properly.

In Eq. (7), epsilon appears in the denominator of the second term in each side, which is not right. Likewise, epsilon in Eq. (15) should be removed. A critical mistake is made in Eq. (12): pf in the denominator of the right hand side should be removed. Meanwhile, in Eq. (14), pf should appear in the denominator of the second term on the right hand side. I doubt that these wrong formulations were actually used in theoretical calculations. If that was the case, the results might have been very different from those shown in the manuscript. The authors should clear this problem.

The authors used a cloud parcel model to calculate the evolution of cloud droplet size distribution during the ascent of a cloud parcel after entrainment and mixing. Very similar but much more sophisticated calculations were already made by Wang et al. (2009). Using a cloud parcel model that incorporates a full CCN spectrum, they calculated the evolution of cloud droplet distribution in an ascending cloud parcel that was mixed with just saturated air in several different proportions. Because the mixed air was just saturated, classification of homogeneous or inhomogeneous mixing was irrelevant. However, during the ascent after mixing, supersaturation of the mixed cloud parcel was readjusted and droplet number concentration and size distribution responded accordingly. Right after mixing, mean droplet diameter was reduced due to the newly activated small droplets from the portion of the just saturated air, but because of reduced droplet number concentration, droplet growth was faster and eventually at some altitude above the mean diameter of the mixed cloud parcel became larger than that of the unmixed cloud parcel. Here the key to the faster droplet growth was due to reduced droplet number concentration and increased supersaturation in the mixed cloud parcel after just saturated air was mixed. Such behavior cannot be resolved when a monodisperse CCN distribution is used as was done in Figs. 1 and 2. But for Fig. 3, a polydisperse CCN distribution was used and the evolution of individual size classes was calculated. Similarly to Wang et al. (2009), I urge the authors to show the variation of supersaturation and activated droplet number concentration and to include them in the discussion of faster growth in the mixed cloud parcel.

Obviously the mixing scenario presented in this manuscript is not likely to occur in exactly the same manner in real clouds. As pointed out by the authors, inhomogeneous mixing may occur instead of homogeneous mixing. Cloud parcels may undergo multiple mixing events not only with entrained environmental air but also with neighboring in-cloud parcels as exemplified in Wang et al. (2009). Some discussion should be made in this perspective.

Line 47: one of 'that' should be removed.

Line 192: Equation 2.1 does not exist. Apparently it is meant to be Equation 7.

Line 320: The fact that more aerosols lead to smaller cloud droplets is simply a fundamental aspect of cloud physics. This does not indicate aerosol indirect effect. The key factor of aerosol indirect effect is the anthropogenic increase of aerosol concentration that leads to increased concentration of smaller cloud droplets. So linking the fact that more aerosols lead to smaller cloud droplets to aerosol indirect effect is not appropriate.

Where is Table 1? I only see Table 2.

Reference Wang et al.: Observations of marine stratocumulus microphysics and implications for processes controlling droplet spectra: Results from the Marine Stratus/Stratocumulus Experiment, Journal of Geophysical Research, 114, D18210, doi: 1029/2008JD011035, 2009.

---

## Author Comment (AC1) · 13 May 2016

**Response to Reviewer 1**

We thank the reviewer for the insightful comments and for noting some important corrections. As detailed in the following, we have revised the paper in accordance with each point. Detailed changes are indicated in the highlighted manuscript uploaded with this response. Here, reviewer comments are in blue text and our responses are in black text.

This paper investigates if cloud parcels that go through a mixing event can produce larger droplets than undisturbed parcels (this is called super-adiabatic growth by the authors). Mixed parcels contain less water and fewer droplets than undisturbed parcels, and therefore droplets there grow faster after a mixing event, although starting from a smaller radius. From thermodynamical considerations, the authors show that super-adiabatic growth is expected for a pristine environment when mixed parcels rise to a given height. This height mostly depends on the thermodynamical properties of the cloud and of the environment, and it is independent of the updraft velocity and of the mixing fraction. This result is tested with a parcel model for different updrafts velocities, different polluted environments and for a polydisperse droplet population.

It has been argued in the past that super-adiabatic droplet growth can help to explain rain formation in warm clouds. The authors are able to quantify this effect in idealized conditions, and I think that their results can be used to estimate the relevance of the mechanism in future studies. For these reasons, I think that the work can be a worthy publication for ACP if the authors answer the next questions.

1) One of the inherent assumptions for the parcel model and for the thermodynamical calculations is that the parcel only mixes once with the cloud-free environment, which means that it never mixes with cloudy air. I would like that the authors discuss this assumption more in detail. Clouds are turbulent and continuously mix (see for example Margaritz et al. 2014), which homogenizes the droplet number concentration. In the example from Figure 1, the parcel has to rise for ~3000 seconds without mixing with other cloud parcels in order to become super adiabatic. It seems unlikely for me to find such a parcel in a real cloud.

This comment is of course correct, and the reviewer is right to request further justification of our idealized approach. As stated in the reviewer comment, a cloud parcel continuously mixes with both cloudy air and the environment air throughout its trajectory. Lagrangian results such as those of Magaritz et al. 2014 and others (e.g., several already cited in the paper, including Cooper et al. 2013, de Lozar and Muessle 2016, Lasher-Trapp et al. 2005, Magaritz et al. 2015, and Naumann and Seifert 2015) have demonstrated some effects of internal mixing, especially due to sedimentation when drizzle is present, and that dilution events often take place repeatedly during parcel ascent. The results presented here do not consider those more realistic conditions, but instead are purposefully designed so as to avoid the complexity of a real cloud and look at the idealized response to a single dilution event. Our motivating philosophy is that if we can understand the 'impulse response' from one mixing event with analytical results, then that understanding can be extended to more complex scenarios. The main purpose of this paper is therefore to study how the cloud microphysical properties in a diluted parcel change when it rises adiabatically after the mixing event, and indeed the derived results are shown to be consistent

with a more detailed (yet still idealized) parcel model. The analytical results might be useful to calculate the entrainment rate profile for the real cloud, similar to Lu et al. (2012), for example. We have added more discussion in the introduction to motivate our idealized approach and how it can be placed in context with more complex, real cloud turbulence. As part of that discussion we include a citation to Magaritz et al. (2014), as well as to several other papers that draw attention to the microphysical effects of internal and external mixing (Korolev et al. 2013, Wang et al. 2009).

2) It would be interesting to know the authors conclusions about the role of super adiabatic droplets for large droplets production and rain formation, from the results presented in the paper. Do they think that the mechanism is relevant for all warm clouds or only in a few particular cases (very wet and very clean environment)? Do they think that the mechanism is relevant for stratocumulus (which are usually thin (~300 m), with a dry capping free atmosphere and with mixing only on the top)? Can they estimate how does the droplet size distribution broaden due to this mechanism (is it sufficient for rain formation)?

Our results show that a mixed/diluted parcel is more likely to reach the superadiabatic growth region if the environmental air is wet and clean. This mechanism might be relevant not only for cumulus, but also for stratocumulus cloud. For example, as mentioned by reviewer 2, Wang et al. (2009) describe a circulation mixing hypothesis to explain microphysical properties in stratocumulus clouds. The circulation mixing hypothesis is similar in spirit to the assumption in our study, except it is conceptually dependent on multiple dilution events and therefore somewhat more qualitative. The circulation hypothesis of Korolev et al. (2013) also has some similarities and is able to produce large droplets that, by our definition would be considered superadiabatic. Our results focus on the possibility of enhanced growth of cloud droplets when they enter the superadiabatic growth region. When those enhanced-growth cloud droplets are mixed with other cloud parcels, the size distribution will be broadened, and while we mentioned this possibility, we have not quantified the broadening effect. This is definitely a key topic that needs to be the focus of future research. We have added some discussion of these points in the introduction and discussion sections.

Technical corrections:

1) Line 120. Equation (4) can be directly obtained from Eq. (1) in the quasi-stationary limit. No need to refer to Eqs. (2) and (3).

Thanks for the reviewer's comment. Yes Equation 4 can be obtained from Equation 1 and we have reordered the equations to make this clear. Because quasi-stationary supersaturation (former Equation 2) is used to explain the enhanced growth of cloud droplet in the mixing parcel, we still keep it.

2) Line 135. There is a prefactor missing in the definition of the liquid potential temperature, which accounts for the pressure dependence. With the current definition, liquid potential temperature is only conserved for adiabatic and isobaric processes. Also, I do not see why \epsilon appears in the definition (it does not appear in Gerber et al. 2008).

Thanks for the reviewer's comment. Yes, we didn't consider the pressure effect on liquid potential temperature. This assumption works if the cloud is not thick. We add more discussion in the text. Because the unit of lw is J/mol (not J/kg) and the unit of cp is J/mol/K (not J/kg/K) (see Appendix), epsilon is a constant to change the unit of lw to J/kg, and change the unit of cp to J/kg/K. Our unit is not as usual as previous paper, but it's consistent in our paper and consistent with that in Lamb and Verlinde's textbook. There were no errors and our results were correct. Since both reviewers were confused by the notation, however, we have changed to the mass-based units in the text since this is the most common terminology.

3) Line 141. Equation number missing.

We've added the equation number in Line 141.

4) Line 151. I would not discuss the state im (after mixing but before phase changes). It is not very useful for the discussion, and it adds more symbols.

We've removed $q_{v,im}$ and $T_{im}$ in the text.

5) Line 156 and 192. Equation 2.1 does not exist.

Equation 2.1 should be Equation 6 (former Equation 7). We've changed it in the text.

6) Line 246. Explain better why ql,fm /ql,f = \chi is the condition for the critical height. Remind the condition of completely clean environment.

This condition only works for the cloud parcel with monodisperse cloud droplet when mixing with clean environment. For a clean environment and homogeneous mixing, it's true that $n_{d,fm}/n_{d,f}$ = \chi as long as droplets in the mixing parcel don't totally evaporate. Here $n_d$ means the cloud droplet number concentration. For monodisperse cloud droplet, if ql,fm/ql,f also equals to \chi, it means that the droplet sizes in both mixing and original parcels are the same. Because ql=4/3pi rhol r^3 Nd. More explanation is added to explain the condition for the critical height.

7) Line 316. Provide the number of CCN in the polluted environment.

For the polluted case, the dry aerosol distribution in the environment is the same as that below the cloud. The total number concentration of aerosol is 50 #/g. The number of cloud droplet can be seen in the new Figures in the supplementary material.

Finally, we have slightly edited the abstract for concision and clarity in accordance with the implemented changes.

---

## Author Comment (AC2) · 13 May 2016

**Response to Reviewer 2**

We thank the reviewer for the insightful comments and for noting some important corrections. As detailed in the following, we have revised the paper in accordance with each point. Detailed changes are indicated in the highlighted manuscript uploaded with this response. Here, reviewer comments are in blue text and our responses are in black text.

In this manuscript the authors tried to tackle the problem of super-adiabatic droplet growth, which has been a subject of great interest in cloud physics community for the last several decades. For without such growth, warm rain initiation within a realistic time scale seems very difficult, if not impossible. The authors considered entrainment and mixing processes as a key to the super-adiabatic droplet growth and derived equations that could calculate analytically the variation of temperature and liquid water mixing ratio and thus droplet radius after entrainment and mixing and during the further ascent of the mixed cloud parcel. Then the authors demonstrated that this theoretical formulation was consistent with the results of cloud parcel model simulations of such processes. Moreover, the authors suggested some proper environmental conditions for super-adiabatic droplet growth. This manuscript does add some new insights on super-adiabatic droplet growth and could be worth publication in ACP if the following issues are handled properly.

In Eq. (7), epsilon appears in the denominator of the second term in each side, which is not right. Likewise, epsilon in Eq. (15) should be removed. A critical mistake is made in Eq. (12): pf in the denominator of the right hand side should be removed. Meanwhile, in Eq. (14), pf should appear in the denominator of the second term on the right hand side. I doubt that these wrong formulations were actually used in theoretical calculations. If that was the case, the results might have been very different from those shown in the manuscript. The authors should clear this problem.

We appreciate that the reviewer has checked our equations carefully. For Eq. (7) (now it's Eq. 6), epsilon in fact should be there because we were using molar rather than mass units: the units of lw are J/mol (not J/kg) and the units of cp are J/mol/K (not J/kg/K). Epsilon is a constant to change the units of lw to J/kg, and to change the units of cp to J/kg/K. Our units are not as typical as in some previous papers, but are consistent within our paper and are consistent with those in Lamb and Verlinde's widely used textbook. There were no errors and our results were correct, but since both reviewers were confused by it, we have changed to the mass-based units in the text since this is the most common terminology.

In Eq. (12) and (14) (now they are Eq. 11 and 13), there are indeed some typos. As the reviewer stated, pf should not be in Eq. (12) but should appear in the denominator of the second term on the right hand side. Our final results (Eq.15-18) are correct, however. The constant C2 is actually from the last term of Eq. (14), and pf exists in C2 (see Appendix).

The authors used a cloud parcel model to calculate the evolution of cloud droplet size distribution during the ascent of a cloud parcel after entrainment and mixing. Very similar but much more sophisticated calculations were already made by Wang et al. (2009). Using a cloud parcel model that incorporates a full CCN spectrum, they calculated the evolution of cloud

droplet distribution in an ascending cloud parcel that was mixed with just saturated air in several different proportions. Because the mixed air was just saturated, classification of homogeneous or inhomogeneous mixing was irrelevant. However, during the ascent after mixing, supersaturation of the mixed cloud parcel was readjusted and droplet number concentration and size distribution responded accordingly. Right after mixing, mean droplet diameter was reduced due to the newly activated small droplets from the portion of the just saturated air, but because of reduced droplet number concentration, droplet growth was faster and eventually at some altitude above the mean diameter of the mixed cloud parcel became larger than that of the unmixed cloud parcel. Here the key to the faster droplet growth was due to reduced droplet number concentration and increased supersaturation in the mixed cloud parcel after just saturated air was mixed. Such behavior cannot be resolved when a monodisperse CCN distribution is used as was done in Figs. 1 and 2. But for Fig. 3, a polydisperse CCN distribution was used and the evolution of individual size classes was calculated. Similarly to Wang et al. (2009), I urge the authors to show the variation of supersaturation and activated droplet number concentration and to include them in the discussion of faster growth in the mixed cloud parcel.

We thank the reviewer for bringing our attention to the very interesting work of Wang et al. (2009). It does have some common aspects and we have added discussion of those aspects to our paper. As requested, we have generated figures that show the variation of supersaturation and the activated droplet number concentration for all cases in our paper. We agree that they are useful to include for those readers who are interested in the details.  Because this would nearly double the number of figures in this short paper and because they are not completely focused on the central message of super-adiabatic droplet growth, we propose that they be included in an online supplement where readers interested in these quantities can easily access them. The variation of supersaturation and activated droplet number concentration profiles are mentioned in the text with a reference to the supplemental figures.

Obviously the mixing scenario presented in this manuscript is not likely to occur in exactly the same manner in real clouds. As pointed out by the authors, inhomogeneous mixing may occur instead of homogeneous mixing. Cloud parcels may undergo multiple mixing events not only with entrained environmental air but also with neighboring in-cloud parcels as exemplified in Wang et al. (2009). Some discussion should be made in this perspective.

Thanks for the reviewer's comment. Reviewer 1 also raised a similar point and we have added a more thorough discussion of mixing with neighboring in-cloud parcels in the discussion section.

Line 47: one of 'that' should be removed.

We've removed one 'that'.

Line 192: Equation 2.1 does not exist. Apparently it is meant to be Equation 7.

We've changed it to Eq. 6 (former Eq. 7).

Line 320: The fact that more aerosols lead to smaller cloud droplets is simply a fundamental aspect of cloud physics. This does not indicate aerosol indirect effect. The key factor of aerosol indirect effect is the anthropogenic increase of aerosol concentration that leads to increased

concentration of smaller cloud droplets. So linking the fact that more aerosols lead to smaller cloud droplets to aerosol indirect effect is not appropriate.

Yes, we agree with the reviewer's point and have removed that sentence.

Where is Table 1? I only see Table 2.

Table 1 was in the Appendix. To avoid any confusion, and because those results are not critical for the main flow of the paper, we now have moved Table 2 to the online supplementary material.

Finally, we have slightly edited the abstract for concision and clarity in accordance with the implemented changes.

---

## Author Comment (AC5) · 13 May 2016

**Supplementary material for "Conditions for super-adiabatic droplet growth after entrainment mixing"**

Fan Yang[1], Raymond Shaw[1] and Huiwen Xue[2]

[1]Atmospheric Sciences Program and Department of Physics, Michigan Technological University, Houghton, Michigan

[2]Department of Atmospheric and Oceanic Sciences, School of Physics, Peking University, Beijing, China

Table S1: Initial dry aerosol radii for different bins.

| Bin number | $r_{dry}$ (nm) | Bin number | $r_{day}$ (nm) |
|---|---|---|---|
| 1 | 463.7 | 11 | 61.5 |
| 2 | 378.9 | 12 | 50.3 |
| 3 | 309.6 | 13 | 41.1 |
| 4 | 253.0 | 14 | 33.6 |
| 5 | 206.7 | 15 | 27.4 |
| 6 | 168.9 | 16 | 22.4 |
| 7 | 138.0 | 17 | 18.3 |
| 8 | 112.8 | 18 | 15.0 |
| 9 | 92.1 | 19 | 12.2 |
| 10 | 75.3 | 20 | 10.0 |

[Figure]

Figure S1: (a) Saturation ratio and (b) cloud droplet number concentration profiles for various cloud mixing fractions when monodisperse cloud droplets mix with clean environment air with low updraft velocity (0.1 ms⁻¹).

[Figure]

Figure S2: (a) Saturation ratio and (b) cloud droplet number concentration profiles for various cloud mixing fractions when monodisperse cloud droplets mix with polluted environment air with low updraft velocity (0.1 ms⁻¹).

[Figure]

Figure S3: (a) Saturation ratio and (b) cloud droplet number concentration profiles for various cloud mixing fractions when monodisperse cloud droplets mix with clean environment air with high updraft velocity (1.0 ms⁻¹).

[Figure]

Figure S4: (a) Saturation ratio and (b) cloud droplet number concentration profiles for various cloud mixing fractions when monodisperse cloud droplets mix with polluted environment air with high updraft velocity (1.0 ms⁻¹).

[Figure]

Figure S5: (a) Saturation ratio and (b) cloud droplet number concentration profiles for various cloud mixing fractions when polydisperse cloud droplets mix with clean environment air with low updraft velocity (0.1 ms$^{-1}$).

[Figure]

Figure S6: (a) Saturation ratio and (b) cloud droplet number concentration profiles for various cloud mixing fractions when polydisperse cloud droplets mix with clean environment air with high updraft velocity (1.0 ms$^{-1}$).